# Effects of Cadmium, Lead, and Mercury on the Structure and Function of Reproductive Organs

**DOI:** 10.3390/toxics8040094

**Published:** 2020-10-29

**Authors:** Peter Massányi, Martin Massányi, Roberto Madeddu, Robert Stawarz, Norbert Lukáč

**Affiliations:** 1Department of Animal Physiology, Slovak University of Agriculture in Nitra, Tr. A. Hlinku 2, SK 94976 Nitra, Slovakia; norbert.lukac@uniag.sk; 2Institute of Biology, Pedagogical University of Kraków, ul. Podchorążych 2, 30-084 Kraków, Poland; robert.stawarz@gmail.com; 3Department of Animal Husbandry, Slovak University of Agriculture in Nitra, Tr. A. Hlinku 2, SK 94976 Nitra, Slovakia; martinmassanyi@yahoo.com; 4Department of Biomedical Sciences-Histology, University of Sassari, Viale San Pietro 43/B, 07100 Sassari, Italy; rmadeddu@uniss.it

**Keywords:** toxic metals, cadmium, lead, mercury, reproduction, testicular and ovarian structure

## Abstract

Reproductive organs are essential not only for the life of an individual but also for the survival and development of the species. The response of reproductive organs to toxic substances differs from that of other target organs, and they may serve as an ideal “barometer” for the deleterious effects of environmental pollution on animal and human health. The incidence of infertility, cancers, and associated maladies has increased in the last fifty years or more, while various anthropogenic activities have released into the environment numerous toxic substances, including cadmium, lead, and mercury. Data from epidemiological studies suggested that environmental exposure to cadmium, lead, and mercury may have produced reproductive and developmental toxicity. The present review focused on experimental studies using rats, mice, avian, and rabbits to demonstrate unambiguously effects of cadmium, lead, or mercury on the structure and function of reproductive organs. In addition, relevant human studies are discussed. The experimental studies reviewed have indicated that the testis and ovary are particularly sensitive to cadmium, lead, and mercury because these organs are distinguished by an intense cellular activity, where vital processes of spermatogenesis, oogenesis, and folliculogenesis occur. In ovaries, manifestation of toxicity induced by cadmium, lead, or mercury included decreased follicular growth, occurrence of follicular atresia, degeneration of the corpus luteum, and alterations in cycle. In testes, toxic effects following exposure to cadmium, lead, or mercury included alterations of seminiferous tubules, testicular stroma, and decrease of spermatozoa count, motility and viability, and aberrant spermatozoa morphology.

## 1. Introduction

Reproduction is an important biological trait to produce new individual organisms and is fundamental for the life of an individual as well as the survival and development of the species [1]. The reproductive system controls the morphological development and physiological differences between males and females as well as influences the behavior of the organism. Environmental and occupational exposure to toxic elements produces various alterations to the biological system, and infertility is one of the global public health concerns as it affects 15% of couples of reproductive age. The toxic mechanisms are described as ion mimicry, disruption of cell signaling pathways, oxidative stress, altered gene expression, epigenetic regulation of gene expression, apoptosis, disruption of the testis–blood barrier, inflammation, and endocrine disruption [2].

Industrial development and agricultural activities have resulted in varying degrees of environmental pollution and reorganization of toxic elements in the food chain [3]. Many elements have been described as highly toxic, while others are essential to living systems. Subfertility and sterility are male reproductive disorders and are interesting in relation to the environment. Adverse trends in male reproductive health are related to environmental impact and have attracted increased attention recently. Declining spermatozoa counts and an increase in reproductive disorders in some areas have been reported during the past 50 years. Because of the relatively short time period in which this has occurred, a crucial role of the environment compared to genetic factors is suggested. Alteration in male reproductive system function may serve as a very sensitive marker of environmental hazards, and the effects can clearly affect reproductive function. The best-reported risk factors related to male reproductive function have included physical exposure and chemical exposure. The doses and exposures demonstrate relatively well the impact of toxic elements at low levels of exposure to the male reproductive structure and function.

Impaired reproductive function is often related to environmental exposure to toxic substances, including toxic metals, namely cadmium, lead, and mercury to which most populations are exposed. These metals are listed by the World Health Organization (WHO) as toxicants of major public health concern (https://www.who.int/ipcs/assessment/public_health/chemicals_phc/en/). Accordingly, we sought to review evidence for reproductive toxicity induced by cadmium, lead, and mercury focusing on changes in the structure and function of male and female reproductive organs of various animal species. Some of these aspects have been covered in published reports [4,5,6,7,8,9]. It was conceivable that male and female reproductive toxicity should be best described by various and new perspectives with respect to dose-dependent effects of environmental exposure on the spermatozoa and ova/follicles [10,11,12,13,14,15]. In addition, we discussed evidence from epidemiological studies that linked impaired reproductive function to cadmium, lead, and mercury to which most people are exposed through their diet [3].

## 2. Cadmium (Cd)

Cadmium is an environmental contaminant from industrial processes and agricultural activities [9,16,17,18,19]. Food is the main source of cadmium exposure for the non-smoking general population [18,19,20]. Cadmium absorption from food in humans is relatively low (3–5%), and it is efficiently reabsorbed in kidneys, with a long biological half-life estimated from 10 to 30 years [21,22,23,24,25,26,27]. In relation to male infertility, cadmium is ranked as a highly toxic element [28,29].

### 2.1. Toxicity of Cadmium in Female Reproductive Organs

The effects of cadmium on the structure and function of ovaries were first reported in 1959 [30]. A significant study was published, a couple of years later, to describe the effect of a subcutaneous injection of cadmium salts on the ovaries of adult rats in persistent estrus [31].

In one of our early experimental works, cadmium was administered to 32 adult mice of the ICR Institute of Cancer Research (ICR) strain in two single doses (0.25 and 0.5 mg CdCl_2_, per kg bw, via i.p. route), and a particularly high cadmium concentration was found in ovaries 48 h after cadmium administration [32]. Similarly, high cadmium accumulation in reproductive organs was observed in an experiment with rabbits [33].

In ovaries of rabbits, a decreased relative volume of growing follicles and increased stroma were found following cadmium administration (i.p. and p.o.) [34,35]. The number of atretic follicles was also significantly elevated by cadmium. The undulation of external nuclear membrane and dilatation of perinuclear cistern and endoplasmic reticulum were the most frequently observed ultrastructural alterations of granulosa, luteal, stroma, and endothelial cells [36]. In all types of cells studied, altered mitochondrial structures were evident [37]. In a later study, the effect of cadmium on the ovarian structure in Japanese quails was studied [38]. A reduction in relative volume of primary follicles was seen in cadmium-treated groups [32]. In addition, the number of follicles undergoing atresia increased as did the number of atretic primary follicles and atretic growing follicles in cadmium-treated groups [38].

In another study, exposed rats had a significantly higher ovarian weight and a higher number of antral and atretic follicles, compared with controls [33]. The effects of cadmium on ovarian follicles were related to changes in gonadotropin hormones and decreases in follicular stimulating hormone (FSH) and luteinizing hormone (LH). A significant formation of oxidative stress in ovarian of Cd-exposed rats was evident from increased levels of the lipid peroxidation product, malondialdehyde (MDA) in combination with decreased levels of an antioxidant enzyme, catalase [39].

In adult female Wistar rats exposed to cadmium, a statistically significant prolongation of the cycle was noted [40]. This effect was observed not only by the estrus phase but also by the diestrus phase. At the highest dose of cadmium (4.5 mg/kg bw), diestrus was extended, whereas proestrus was shortened although no significant increase in the duration of the cycle length was observed. Histopathological alterations in the ovaries of the cadmium-exposed rats were degeneration of the corpus luteum and damaged and fewer oocytes [40].

Cadmium decreases antioxidants and increases the concentrations of MDA and hydrogen peroxide (H_2_O_2_) in ovaries of rats [41]. Histopathological analysis of the ovaries indicates a significant decrease in follicle number [41]. Ovaries of the cadmium-exposed rats showed a decrease in the number of follicles, with a distorted Graafian follicle.

Cadmium also affects the maturation of follicles, degradation of the corpus luteum and the arrangement of follicles and corpus luteum and increases the number of atresia follicles [42]. In an analysis of ovary morphology, visualized by the expression of the granulosa-cell-specific factor (AMH), there were no differences in the follicle number in each stage (primordial, primary, secondary, and antral follicles) in cadmium-exposed groups [43]. The number of TUNEL-positive cells in the ovaries of cadmium-exposed groups did not differ from controls as did the levels of SOD and MDA in the ovaries [44].

An age-specific effect of cadmium in women was evident from a study using anti-Mullerian hormone (AMH) that is secreted by granulosa cells of antral follicles as an indicator of ovarian function [44]. An inverse association between cadmium and AMH was seen only in women aged between 30 and 35 years. Therefore, it is concluded that environmental exposure to cadmium may alter the AMH level and ovarian function depending on age [44].

Exposure to moderate and high doses of cadmium (i.p.; 1 mg/kg for 5 days/week for 6 weeks) affects steroid synthesis in reproductive organs in female rats [45]. A low-dose cadmium exposure has potent estrogen- and androgen-like activities as cadmium directly binds to estrogen and androgen receptors [46]. Cadmium, like estradiol, can cause rapid activation of ERK1/2 and AKT [47]. However, the exact mechanisms explaining cadmium as an endocrine disruptor remain to be investigated.

The toxicity of cadmium on female reproduction was studied in birds (50-day-old Hy-Line white hens, fed with two doses containing 140 and 210 mg/kg CdCl_2_ for 20, 40 and 60 days), and increased MDA, nitric oxide (NO) and the activity of nitric oxide synthase (NOS) in the ovary were noted [48]. In addition, the levels of glutathione peroxidase (GPx) and SOD activity decreased in experimental groups. The number of apoptotic cells in the ovary increased in the cadmium-exposed groups, and extensive damage was observed in the ovary [49].

Table 1 summarizes the most significant changes in the ovary due to cadmium exposure.

A few studies have examined effects of cadmium on the structure and function of the oviduct. The highest relative volume of epithelium was seen in the oviduct after long-term Cd administration [36]. Histological analysis reports edematization of the oviduct tissue, which is related to the disintegration of the capillary wall. Further analysis describes dilatation of perinuclear cistern, enlarged intercellular spaces and alterations in cell junctions. Mainly after a long-term cadmium administration, nuclear chromatin disintegration was present [36]. The effects of cadmium on embryo transport through the oviduct were studied in the rat after administration of 2.5, 5, 10 mg/kg CdCl_2_ on day 1 of pregnancy. Cadmium accumulated in oviducts in dose- and time-dependent manners [49]. Our previous studies reported similarly high concentrations of cadmium in the oviduct [50,51].

Effects of a single subcutaneous injection (1.5 mg/100 g bw) of cadmium chloride were studied in the oviduct of Indian koel (*Eudynamys scolopacea*). The stromal tissue showed hyperemia and profuse hemorrhage, leading to some cellular destruction. A marked degeneration was noticed in the lamina propria of magnum [53].

A large population-based cohort study of Swedish postmenopausal women described an association between dietary cadmium intake and endometrial cancer and showed potential estrogenic effects [54]. The average estimated cadmium intake was 15 μg/day and 378 cases of endometroid adenocarcinoma were found in a 16-year follow-up. It is noteworthy that an average cadmium intake level among Swedish women of 15 μg/day was 25.8% of an established safe intake level of 58 µg/day for a 70-kg person (0.83 μg/kg bw per week) [18].

Increased incidence of hormone-related cancers and diseases in Western populations may reflect that cadmium has estrogen-like activity in vivo [55]. Environmental contaminants that mimic the effects of estrogen have been linked to the disruption of the reproductive systems of wild animals [55].

A morphological study during the estrus stage reported that cadmium administered orally (0.09, 0.9, 1.8, and 4.5 mg/kg bw) for 90 days caused increased thickness of the epithelial layer. However, in high doses, cadmium induces atrophy of endometrium [55]. Authors also conclude that high-dose cadmium does not induce estradiol-like hyperplasia of endometrium but results in endometrial edema [55].

In the rabbit uterus, a significant adverse effect of cadmium exposure was reflected by the relative volume of glandular epithelium [36]. The increase of stroma was a sign of uterus edematization caused by damage in the wall of blood vessels and subsequent diapedesis [36].

In a study examining myometrial responsiveness, cadmium exposure increased both absolute tension and mean integral tension in female rats exposed to 3, 10, and 30 ppm of cadmium in drinking water for 28 days [56]. Cadmium accumulated in the myometrium of rats and altered response to oxytocin, histamine, calcium chloride, and phenylephrine. These effects were differentially mediated depending on the levels of exposure, possibly through voltage-dependent calcium channel and Ca^2+^-mimicking pathways [57]. In another study, female BALB/c mice were exposed to 200 ppm cadmium in drinking water for either 30 or 60 days. Cadmium exposure resulted in significant decreases in endometrial thickness, number of glands in estrus-phase uteri and endometrial eosinophilia. Cadmium exposure also increased the number of mast cells. The apoptotic index increased with time in both experimental cadmium-exposed groups, while the proliferation index decreased. Authors conclude that 60-day Cd exposure increased apoptosis in the endometrium, which may affect the receptivity of the uterus for implantation [58].

Cadmium contents in uterine cancer and uterine myoma were studied, and a correlation was observed between tissue cadmium contents and age [59]. An analysis of cadmium content of collected samples of uterine myomas, uterine cancer, and non-lesion uterine tissues from the same women, aged 32–79 showed that cadmium content of myoma was lower than non-lesion tissue [60].

### 2.2. Toxicity of Cadmium in Male Reproductive Organs

For cadmium as a toxic element, people and animals are exposed to it through contaminated food and the environment. It has been reported that cadmium causes damage to the male testis in animals as well as humans. Cadmium was stated to induce alterations of seminiferous tubules, Sertoli cells, blood–testis barrier, and the loss of spermatozoa. Cadmium alters Leydig cell development and function and induces Leydig cell tumors. Furthermore, cadmium disrupts the vascular system of the testis. As an inducer of reactive oxygen species, cadmium possibly causes DNA damage and subsequently leads to male subfertility/infertility [61].

In a study with two experimental groups exposed to cadmium via injection: acute exposure to CdCl_2_ 3 mg/kg for 5 days, and chronic exposure to CdCl_2_ 1 mg/kg for 30 days. In acute exposure conditions, part of seminiferous tubes swelled and the arrangement of spermatogenic cells was mildly disordered [62]. In a chronic exposure conditions, the layers of seminiferous tubes were irregular, and there was loss of spermatogenic cells and a dramatic increase of spermatozoa in tubules [63].

In rats receiving subcutaneous injections of CdCl_2_ (3 mg/kg bw) once a week for four weeks, cadmium induced biochemical alterations in testicular tissues such as increase in MDA and decrease in antioxidant markers SOD, catalase (CAT), and glutathione (GSH) and functional markers such as alkaline phosphatase (ALP) and lactate dehydrogenase (LDH) [63]. Microscopic changes were manifested as desquamation of basal lamina, shrunken tubules, generalized germ cell depletion with multinucleated giant cells, degenerating Leydig cells, vascular congestion, interstitial edema, and a significant reduction in spermatodynamic count [63].

Tubular damage (vacuolization of the seminiferous epithelium, germ cell detachment and seminiferous tubule degeneration) was seen in the Swiss adult male mice received CdCl_2_ (1.5 mg/kg i.p., 30 mg/kg oral single dose, and 4.28 mg/kg oral fractional dose for 7 consecutive days) [63]. Authors observed also seminiferous epithelium degeneration, death of germ cells and Leydig cell damages, which were evident in both groups of mice receiving cadmium via i.p. and oral routes. Of note, the damages were more intense in the oral route than i.p. route [64]. In another experiment, the reduction in the diameter of seminiferous tubules was also found [65].

In a study aimed at the mechanisms of cadmium-induced reproductive toxicity in a male mouse model, results demonstrated that the severity of testes injury increased with cadmium concentrations [66]. In low doses, cadmium decreased the thickness of the testicular seminiferous tubule walls, with less apparent swirling contours, and no obvious changes in the appearance of testicular stromal cells were detected. In moderate doses, cadmium caused the thinning of germinal epithelium, sporadic bleeding in the testicular stroma, cells with aberrant swirling contours and decreased spermatogenesis. In high doses, cadmium caused severe thinning of germinal epithelium, seminiferous tubules with aberrant morphology, markedly low level of normal spermatogenesis, and apparent abnormalities of the testicular stroma [66].

In subchronic exposure conditions using rats, structural changes in testis and epididymis were observed together with the increases in testis and epididymis weights 90 days after peroral administration of cadmium at 30 mg/L of drinking water [67]. Testicular damage seen included a significant thickening of seminiferous epithelium, cellular degeneration, and necrosis [60]. Desquamation of immature germ cells resulted in a significant increment of intraepithelial spaces and reduction of tubular volume [67].

Table 2 summarizes the most significant changes in testes due to cadmium exposure.

Activation of the Erk/MAPK signaling pathway has been proposed as a driver for cadmium-induced prostate cancer. In an in vitro study using non-malignant human normal prostate epithelial cells and PWR1E cell line, chronic exposure to cadmium caused a significant increase in cell proliferation in conjunction with a reduction in apoptosis [69]. An increase in phosphorylation of the Erk1/2 and Mek1/2 was observed in Cd-RWPE1 and Cd-PWR1E cells compared to parental cells, thereby confirming that Cd-exposure induces activation of the Erk/MAPK pathway. It is concluded that Erk/MAPK signaling may be involved in malignant transformation of normal prostate cells by cadmium [70].

Chronic cadmium exposure causes also defective autophagy in prostate epithelial cells, leading to a malignant phenotype transformation [71]. Authors suggest also that chronic exposure to cadmium induces endoplasmic reticulum (ER) stress, which triggers the phosphorylation of stress transducers, and results in defective autophagy in RWPE-1 cells following exposure to cadmium [68].

## 3. Lead (Pb)

Lead is used in lead acid batteries, coloring agents, paints, smelters, and printing presses and is metallically alloyed as shielding material. It is a toxic metal affecting various organs and developing fetus [72,73]. Acute lead poisoning occurs in humans exposed to high doses, and its chronic exposure can become fatal when lead builds up in the body gradually via continuous exposures to small amounts [74]. It affects almost all organs of the human body and causes physical and mental impairments. Lead remains a noteworthy cause of environmental, occupational, public, and animal health problems [75,76,77,78,79].

### 3.1. Toxicity of Lead in Female Reproductive Organs

In 1960, a paper describing alteration in the ovary of the rhesus monkey with chronic lead intoxication was published [80], and authors stated that it has long been known that women exposed to lead show disturbances in menstruation, such as amenorrhea, dysmenorrhea, and menorrhagia. Later, results of an experiment with rats (control vs. treated with 5 µg or 100 µg for 30 days) indicated irregularity of the estrus cycle, while the group treated with higher concentrations had persistent vaginal estrus after a period of normal estrus, and the development of ovarian follicular cysts with a reduction in the number of corpora lutea was noted [81].

Lead exposure affects female reproduction mainly by impairing menstruation, reducing fertility potential, delaying conception time and altering hormonal production and circulation, affecting pregnancy and its outcome [82]. Reported effects of lead include infertility, miscarriage, early membrane rupture, preeclampsia, pregnancy hypertension, and preterm birth [83]. A review of studies from China described the possible links between low-level lead exposure and adverse effects on the reproductive system. Effects manifested mainly as high prevalence rates of menstrual disturbance, spontaneous abortion, and threatened abortion in exposed females [84].

Lead exposure is associated with hormonal imbalance causing reproductive impairment, and the accumulation of lead affects many endocrine glands [85]. It affects the hypothalamic–pituitary axis, causing blunted thyroid-stimulating hormone, growth hormone, and follicle-stimulating hormone (FSH)/luteinizing hormone (LH) responses to thyrotropin-releasing hormone, growth hormone–releasing hormone and gonadotropin-releasing hormone stimulation.

A study was done on female workers with a mean age of 32 years employed in a storage battery with a lead exposure period of 7.4 years (1–17 years). The incidence of polymenorrhea, prolonged and abnormal menstruations, and hypermenorrhea was significantly increased in the lead-exposed group, and the incidence of spontaneous abortions was also reported. Authors concluded that the occupational lead exposure results in impairment of the reproductive functions [86].

In a study of 259 healthy women, aged 18–44 years, the amplitude of estradiol showed a tendency to decrease with increasing lead exposure [87]. Thus, environmental exposure to lead experienced by healthy premenopausal women may produce modest changes in reproductive hormone levels [88].

Increment of serum FSH levels with increasing blood lead levels was observed in a study of women, aged 35–60 years [89]. Such increase of serum FSH levels with blood lead levels was seen in postmenopausal women, women with both ovaries removed and premenopausal women [89]. These data suggest that lead may act directly or indirectly to increase ovarian concentrations of FSH and LH [90].

High blood lead levels were associated with a delay in the onset of puberty, after adjustment for possible confounders [89]. Blood lead levels (≥5 μg/dL) were associated with lower levels of maturation at 9 years of age, and slower progression of pubic hair and breast development [91]. Gender differences in the effects of prenatal and postnatal lead exposure on pubertal development have been noted [90]. Accumulating evidence suggests that chronic exposure to low levels of environmental lead may have the following effects in females: menstruation cycles, offspring development, the intellectual ability, offspring weight, and hormonal production [91].

Effects of lead accumulation in the ovary and damage to folliculogenesis were seen in mice exposed to lead as PbNO_3_ via intraperitoneal (i.p.) injection at 10 mg/kg/day for 15 days or 10 mg/kg/week for 15 weeks [92]. Dysfunction of folliculogenesis, expressed as decreased primordial follicles and an increase of atretic antral follicles was observed [88]. No significant difference in antral follicles was found. The percentages of primordial follicles were significantly lower than in controls (39.7 ± 3.5% vs. 50.7 ± 3.2%), while the percentages of growing follicles (44.7 ± 4.5% vs. 35.0 ± 5.3%) and atretic follicles (17.2 ± 2.4% vs. 4.6 ± 0.8%) were significantly higher compared to control ovaries. In lead-exposed mice, more oocytes had resumed meiosis in the follicles compared to controls. The ovaries from lead-exposed mice contained atretic antral follicles, with detached granulosa cells, pyknotic nuclei in the granulosa wall, and a hypertrophic theca layer.

Another study observed the effects of lead acetate on the histomorphology of the ovary in an animal model—mice of BALBc strain [93]. Animals were given lead acetate at a dose of 30 mg/kg per day for two months. The primary follicular count decreased significantly in the lead-exposed group. In general, the authors also reported that the morphology of the ovary was affected after exposure to lead acetate. Thus, lead clearly interferes with the development of growing follicles [94].

In a study of adult female rats with an oral chronic dose of 60 mg/kg for 90 days, the effect on the reproductive functions was noted [94]. Histological studies of ovaries showed atresia in all the stages of folliculogenesis, supportive of the poor fertility observations. Authors described various stages of follicles undergoing atresia, mainly antrum-formed Graafian follicles with complete detachment of granulosa cells from theca [94].

Histological changes in the ovaries were investigated using Wistar female rats exposed to lead acetate in drinking water in concentration between 0.050 and 0.150 mg/L for 12 months [95]. Such long-term exposure to lead resulted in a statistically significant increase of lead concentrations in ovaries (+42.13% to 9.4-fold increment). In low-dose exposure conditions, some areas with optical empty spaces were present in ovaries with diffuse edemas and ovarian follicle denudation. In high-dose exposure conditions, the most noticeable alterations—edemas and necrosis of the ovarian follicles were observed.

One study compared lead contents of malignant epithelial ovarian cancers (*n* = 20), borderline ovarian tumors (*n* = 15), and non-neoplastic ovarian tissues (*n* = 20) [96]. Lead contents were significantly higher in malignant tissues than controls. Lead contents of papillary and capsular samples of the borderline tumors were also higher than normal ovarian tissues. Thus, accumulation of lead in ovarian tissue showed associations with borderline and malignant proliferation of the surface epithelium [96].

An inverse association was observed between the amount of lead in follicular fluid from a single follicle and in vitro oocyte fertilization outcomes [97]. Later, a higher level of lead was found in the follicular fluid of the women from Taranto compared to the control group. It is concluded that chronic exposure to heavy metals, lead in particular, may decrease the production of estradiol and the number of retrieved mature oocytes [98]. A significantly higher lead concentration was also reported in the polycystic ovary syndrome (PCOS) group compared to the control [99].

Table 3 summarizes the most significant changes in ovaries due to lead exposure. 

An in vitro study of porcine ovarian granulosa cells, low doses of lead (0.063 mg/mL and 0.046 mg/mL) inhibited IGF-I release from ovarian granulosa cells [101]. The authors concluded that lead can affect the pathway of proliferation and apoptosis of porcine ovarian granulosa cells through intracellular substances such as cyclin B1 and caspase 3 [96]. In another in vitro study (with lead acetate), effects of lead on total antioxidant status (TAS) and activity of SOD in ovarian granulosa cells were observed [102]. Further research demonstrated that lead causes a significant reduction in gonadotropin binding, which altered the steroidogenic enzyme activity of granulosa cells [103,104,105,106].

The reported effects of lead acetate (10 mg/kg) on uterus histomorphology include lead-induced inflammatory alterations, which were characterized by narrowing of the uterine lumen; atrophy of the endometrium; vacuolar degeneration in endometrial epithelial cells; damaged and decreased number of endometrial glands; and increased filtration of inflammatory cells [105]. In a previously described experiment [95], the levels of lead in uterus with oviduct were also significantly increased in the experimental groups. At the same level of lead exposure, necrosis of the uterine glands in the uterus and fallopian tubes was detected [95].

### 3.2. Toxicity of Lead in Male Reproductive Organs

Several pathways might be involved in lead-induced impairments of male reproductive health [83,84,105]. Lead reduces male fertility by decreasing spermatozoa quality. The blood–testis barrier can protect testicular tissue from direct exposure to high blood lead concentrations. Furthermore, it has been stated that environmental and occupational exposure to lead may adversely affect the hypothalamic–pituitary–testicular axis, impairing spermatogenesis. Dysfunction at the reproductive axis, mainly testosterone suppression, is most susceptible and irreversible during pubertal development. Lead poisoning also affects the process of spermatogenesis and spermatozoa function. Generation of excessive reactive oxygen species due to lead exposure potentially affects spermatozoa viability, motility, DNA fragmentation, and chemotaxis for spermatozoa–oocyte fusion, all of which can contribute to deterring fertilization [106].

Several reports indicate that lead has toxic effects on male reproduction through libido decline, spermatogenesis, semen quality, and hormonal production and regulation. It has been generally concluded that exposure to low-to-moderate levels of environmental lead affects certain reproductive parameters [107]. Blood lead levels >40 μg/dL have been linked to impaired male reproductive function, possibly by reducing spermatozoa count, volume, and density or by changing spermatozoa motility and morphology [108]. Total spermatozoa count decreases as blood lead increases, and the concentrations of lead in semen show an inverse association with total spermatozoa count, ejaculate volume, and serum testosterone [109]. No significant association was seen between blood lead levels and spermatozoa concentrations [109]. A significant decrease in spermatozoa motility and an increase in testosterone level were seen in patients with a blood lead concentration of ≥20 μg/dL [110].

Various signs of damage in the testicular architecture were noted in adult male Kunming mice (8 weeks old) receiving lead (lead acetate, 100 mg/kg) for 3 weeks [111]. The seminiferous tubules showed disorganization, and tubules were shrunken and distorted with complete absences of the spermatogenesis process. Histological evaluation was further confirmed by the Johnson score, which was lower in the lead-exposed group compared to control [111].

Another study described the effects of lead on the hypothalamic–pituitary–testicular axis, steroidogenesis, spermatozoa parameters, and testicular antioxidant enzyme activity of male Wistar rats after an administration of lead acetate [112]. In this study, adult male Wistar rats received lead (10 mg/kg) had decreased serum luteinizing hormone (LH) and testosterone levels; testicular 17β-hydroxysteroid dehydrogenase (HSD) activity; androgen receptor expression; spermatozoa motility, viability and counts; catalase activity (CAT); and SOD when compared with controls. Abnormal spermatozoa morphology and MDA increased significantly in the lead experimental group [112].

Administration of lead acetate (orally, 20 mg/kg bw, 10 days) to adult rats induced oxidative stress via attenuation of LH, total testosterone, and follicle-stimulating hormone levels in serum [113]. The testes of animals in experimental groups showed an increase in ROS levels, lipid peroxide levels, and lysosomal enzyme activity. The testicular tissue showed clear degeneration with loss of spermatogenic series related to lead administration [114].

The results of another study showed non-significant changes in the absolute and relative weights of epididymis and testes in the lead-exposed group compared with the control, but significant increases were recorded in the spermatozoa analysis and luteinizing hormone, as was a decrease in follicle-stimulating hormone (FSH) [115]. A 25% decrease in testicular CAT activities in the lead-exposed group was detected. The histopathological analysis of testes treated with lead showed edema, hydrocele, and inflamed tunica albuginea [115].

A significant decrease in the weights of testes and epididymis compared to the control group was noted in a study of male albino rats receiving lead acetate (20 mg/kg, orally) for 56 successive days [116]. Decreases in epididymal spermatozoa concentration, motility and viability were also observed. Histopathological analysis showed marked testicular lesions as disorganization and complete hyalinization, tubular blockage with sloughed germinal epithelium and germinal epithelium hypocellularity. The spermatogenesis score showed a significant reduction. Bax antigen staining showed higher intensity in testes of the lead-exposed group [107]. In addition, in lead-exposed rats, strong staining intensity of caspase-3 antigen in Sertoli cells and the resident germ cells was found. Lead induced a significant increase in testicular MDA and NO compared to the control as well as a significant reduction in testicular SOD and GSH [114].

Wistar rats exposed to lead acetate in drinking water for 45 days showed a significant reduction in testis weight, spermatozoa count, testosterone levels, and antioxidant enzymes levels. Testicular histological sections in lead-exposed animals were devoid of germ cells and maturation arrest with the formation of giant primary spermatocytes [116].

Table 4 summarizes the most significant changes in testes due to lead exposure.

In our previous study, we observed the dilatation of blood capillaries in the interstitium, undulation of basal membrane, and occurrence of empty spaces in the seminiferous epithelium in testes of rats that received lead (PbNO_3_) in single intraperitoneal doses (12.5–50 mg/kg) [37]. There was also an increased incidence of apoptosis in the spermatogenetic cells. Further morphometric analysis confirmed significant differences between control and treated groups in evaluated parameters. The number of cell nuclei was decreased in lead-treated groups consistent with the occurrence of empty spaces and higher apoptosis incidence in the germinal epithelium [37,118].

The relationship between semen quality and the concentrations of lead in seminal plasma of various species have been reported in our previous studies [119,120,121,122,123,124]. It has been reported that environmental lead affects semen quality and spermatozoa chromatin considering lead in seminal fluid, spermatozoa and blood as exposure biomarkers in urban men (9.3 μg/dL blood lead). Authors report that 44% of subjects showed decreases in spermatozoa quality, concentration, motility, morphology, and viability that were associated negatively with lead concentration in seminal plasma [121]. Seminal lead concentrations showed inverse associations with spermatozoa concentration, motility, and percentage of abnormal spermatozoa [125]. Authors concluded that exposure to lead (5.29–7.25 μg/dL) affects semen quality [126]. Similarly, blood plasma lead concentrations were reported to be significantly higher in azospermic and oligospermic men compared to normospermic men [127].

## 4. Mercury (Hg)

Mercury exists as elemental, inorganic, and organic forms [128]. Human exposure to mercury occurs mostly through seafood or sashimi consumption, and also to a lesser extent through dental amalgams, broken thermometers, fluorescent light bulbs, button cell batteries, and skin-lightening creams [129,130]. Mercury exposure is related to its concentrations in the environment [131,132,133] and various food products [134,135,136,137,138,139,140]. Possible health impacts have been reported [141].

In an analysis of data from the U.S. National Health and Nutrition Examination Survey (NHANES) 2011–2012 (*n* = 7920), the overall population mean for whole blood total mercury (THg) was 0.70 μg/L [142]. In a comparative study, 3.8% of seafood consumers had whole blood THg higher than 5.8 μg/L, while 9.4% of them had whole blood THg higher than 3.4 μg/L [143]. In addition, seafood consumers had higher geometric mean for whole blood THg (0.89 µg/L) than non-seafood consumers (0.31 µg/L) [143]. Fish/seafood was found to be likely sources of mercury exposure among seafood consumers, whereas wine, rice, vegetables, vegetable oil, or liquor were dietary sources of mercury exposure among non-seafood consumers [142]. Intriguingly, a 2.57-fold increase in risk of infertility was seen among women enrolled in NHANES 2013–2016 who had high levels of whole blood THg (>5.278 μg/L) [143].

### 4.1. Toxicity of Mercury in Female Reproductive Organs

Studies related to the reproductive toxicity of mercury were first reported many years ago [144]. Later studies developed the knowledge of mercury effects on female reproductive organs [145,146]. There are few studies reporting the effect of mercury in animal and human female reproductive organs.

In females, mercury can accumulate in ovaries and can cause changes in reproductive behavior, infertility and ovarian failure [147,148]. Studies using experimental animals have shown that increased doses of mercury inflate the potential number of reproductive disorders (i.e., infertility, stillbirth, congenital malformations, and spontaneous abortion) [149].

Evidence from human and animal studies suggests that mercury may adversely affect reproductive function. Increased mercury levels have been linked to infertility or subfertility. Infertile subjects with unclear reasons for infertility had higher mercury concentrations in hair, blood, and urine than fertile subjects. Mercury exposure has been associated with increased incidence of menstrual and hormonal disorders as well as increased rates of adverse reproductive outcomes [150]. Increased exposure to mercury was seen among subjects with polycystic ovary syndrome (PCOS) in a case-control study of 84 patients with PCOS and 70 healthy volunteers [151]. In females, mercury has been shown to have an inhibitory effect on the release of luteinizing hormone (LH) and follicle-stimulating hormone (FSH) from the anterior pituitary [152,153]. This can tip the levels of estrogen and progesterone, causing painful or irregular menstruation, tipped uterus, premature menopause, and often different ovarian dysfunctions. Mercury exposure has been associated with multiple menstrual disorders such as shortening or prolonging of menstrual cycles, abnormal bleeding, or pain [153].

Female mice exposed to 0.25–1.00 mg/kg/day (gavage) of methylmercury showed reduced fertility and survival indices in experimental groups although the exposure did not affect their litter size. Notably, authors stated that microscopic lesions were randomly distributed among the control and experimental groups, and the severity, as well as the distribution, was not consistent with methylmercury-induced toxicity [154].

A positive relationship was observed between the accumulation of mercury in ovaries and the follicular atresia rate in laying hens (40-week-old Hy-Line Brown) fed with four experimental diets containing graded levels of mercury (0.280, 3.325, 9.415, and 27.240 mg/kg) [155]. Progesterone levels significantly decreased in all mercury treatment groups. On the other hand, FSH and LH levels showed inverse correlations with mercury doses. The activities of catalase, superoxide dismutase, glutathione reductase, and glutathione content significantly decreased in experimental groups [155].

Exposure concentration-related effects of mercury vapor were evaluated in rats exposed to 0, 1, 2, or 4 mg/m^3^ Hg degrees vapor for 2 h/day for 11 consecutive days. Slightly prolonged estrous cycles were detected in groups with higher mercury exposure. A lengthening of the cycle was detected, and morphological changes were observed in the corpora lutea after exposure for 6 days. Authors concluded that exposure to mercury vapor alters the estrous cycle but has no significant effect on ovulation, implantation, or maintenance of first pregnancy [156]. Another study confirms the effect of mercury vapor (HgO) inhalation on rat ovary structure and function. Ovaries exposed to mercury had various histo-morphometric alterations. A reduction of the total number of primordial, primary, and Graaf follicles was detected. The average mean volume of ovary, medulla and cortex, corpus luteum and Graaf follicles was decreased in the mercury-exposed group. There was also a significant increase in the total volume of the atretic follicles [156].

In an experiment (daily s.c. dose; 1 mg of HgCl_2_), it was reported that, in the ovary, more mercury was concentrated in the corpora lutea than in the follicles and interstitium. Moreover, when hamsters were given a total of 3 or 4 mg of HgCl_2_ during the first cycle, 60% of the animals did not ovulate by day 1 of the third cycle [157].

Mercury-exposed rats exhibited irregular estrous cycles and abnormal duration spent in the different phases of the estrous cycle. Mercury-exposed rats (i.m.) spent less time in the proestrus phase compared with control. Authors observed significantly lower ovarian weight in mercury-exposed rats than in control, and ovaries displayed irregular ovarian follicular development. Ovaries had a reduced ovarian antral follicles number and an increase in the number of atretic ovarian follicles (approx. 57%). On the other hand, experimental rats showed an increase in the number of ovarian cystic follicles. No significant changes were observed in the number of primordial, primary, and preantral ovarian follicles and corpus luteum (CL). An increase in the O_2_- levels was also observed in mercury-exposed rats [158].

A statistically significant increase in pre- and early post-implantation fetal losses was confirmed in female BALB/c mice treated with a single intraperitoneal injection (2.5–7.5 mg methyl mercury chloride per kg body weight) [159].

It has been reported that that mercury chloride (1.5 mg/kg bw) damages the ovary function and reduces the number of superovulation oocytes in vitro. Results also show that mercury inhibits the extruding of the first polar body and affects the quality and viability of mouse oocyte. Authors also indicate that mercury could affect the meiotic maturation of mouse oocyte, obviously block the IVF and injure or reduce mice’s reproductive capacity [160].

Table 5 summarizes the most significant changes in ovaries due to mercury exposure.

In an in vitro study of porcine ovarian granulosa, the effect of mercury on secretion activity of progesterone and insulin-like growth factor-I (IGF-I) was analyzed. Results show that progesterone release by granulosa cells was significantly inhibited, while IGF-I release was not affected. An increasing trend of apoptosis of granulosa cells was confirmed. Results of this study also confirm a direct effect of mercury on the release of steroid hormone progesterone as well as interference of mercury in the pathways of steroidogenesis and apoptosis [161]. Other data confirm that mercury administration has a dose-dependent association with the hormonal release by porcine ovarian granulosa cells [162].

Another study analyzed skin-lightening creams, which are used by women in particular in an attempt to whiten their skin, and men and older people use these creams to remove age spots or other pigmentation disorders [148]. The authors found the presence of high mercury levels in skin-lightening cream. The mercury content in the ovary tissues increased with the frequency of cream applications and was highest in the ovaries of mice treated twice a day with Fair & Lovely and once a day with Rose. Data indicate that dermal exposure to mercury can result in significant accumulation in ovaries of mice following skin-lightening cream application [148].

Effects of mercury on oviducts are studied mostly in female birds as the oviduct is important for egg development. A significant eggshell thinning and deformation and inhibited egg production have been noted in domestic fowls receiving methyl mercury—5 mg daily for 6 consecutive days and 1 mg daily for 50 consecutive days [164]. In an in vitro analysis, mercury added to a homogenate of eggshell gland mucosa significantly stimulated the synthesis of prostaglandin F2-alpha (PGF2-α) and prostaglandin E2 (PGE2). Authors conclude that this effect is related to a direct inhibitory effect of mercury on calcium uptake from the gastrointestinal tract and/or to mobilization of medullary bone [164].

The ultrastructure of the surface epithelium of the oviduct of ducks has been reported. Pekin ducks were fed with different doses of methylmercury (0.5; 5.0; 15.0 ppm) for 12 weeks [165]. The primary and secondary folds of magnum and the shell gland regions were densely populated with ciliated cells. In the medium-exposure group, areas of ciliary loss were observed. In the group with the highest mercury exposure, ciliary loss was more extensive and disruption of the apex of cells was also detected. Transmission electron microscopy (TEM) showed degeneration of cytoplasmic organelles—severely damaged ciliated cells, loss of ciliary extensions, and formation of compound cilia [165].

Subcutaneous injection of female hamsters with 6.2–8.2 mg of mercury per kilogram body weight led to a disruption of estrus 1–4 days after treatment [166]. Normal uterine hypertrophy and follicular maturation inhibition, morphological prolongation of corpora lutea, and alteration of progesterone concentrations were noted [167].

Effects of mercury on myometrial activity were examined using Wistar rats receiving 5, 50, and 500 μg/L mercuric chloride in drinking water for 28 days [168]. A significant increase in the receptor-dependent (PGF2α-induced) and receptor-independent (CaCl_2_-induced and high K+-depolarizing solution-induced) myometrial contraction was observed in rats that received the lowest mercury. Authors concluded that mercury at a low dose produced a detrimental effect on myometrial activity by altering calcium entry into the smooth muscle and/or the release of calcium from intracellular stores [167]. These authors also confirmed that mercury (HgCl_2_) produced a concentration-dependent uterotonic effect. This study shows that mercury evidently interacts with muscarinic receptors and activates calcium signaling cascades involving calcium channels, Rho-kinase, protein kinase-C, and phospholipase-C pathways to exert an uterotonic effect in rats [169].

Animals exposed to mercury showed uterus inflammatory cells in the endometrium and myometrium [170]. The uterine endometrial area was decreased compared to control, while the myometrium area showed no difference between the groups [170].

In another human study, elevated mercury concentrations were observed in the hyperplasia tissue samples from pre-menopausal women, aged less than 50 years, who had different endometrial pathologies: typical endometrial hyperplasia, endometrial cancer and normal endometrial tissues [169].

### 4.2. Toxicity of Mercury in Male Reproductive Organs

Negative effects of mercury on reproductive traits in male rats include impairment of spermatogenesis, decrease in spermatozoa motility, and increase in pathological changes [171,172,173,174,175,176,177].

Testicular toxicity of mercury chloride (HgCl_2_) in adult male Wistar rats was reported [174]. In the mercury-exposed group (40 mg/kg bw; HgCl_2_; orally daily; 28 days), histological profiles of the testes showed a derangement of the cytoarchitecture and deterioration of spermatozoa quality. The weights of testes and the gonadosomatic indexes were significantly lower in the mercury-treated group compared to the control. In the mercury-exposed group, degeneration of the spermatogenic cells of the germinal epithelium, occlusion of the lumen of seminiferous tubules, hypertrophy of seminiferous tubules, and irregular vacuolized basement membrane were found. After 28 days of administration, a significant decrease in mean spermatozoa motility and spermatozoa count was detected in the mercury-exposed group [174]. Similar changes in testes were also confirmed after peroral mercury administration. Authors found a decline in spermatozoa, disorganization and degeneration of some spermatogenic cells and vacuolated areas within the seminiferous tubules in the mercury experimental group [175].

Necrosis, disintegration of spermatocytes from basement membrane, undulation of basal membrane and severe edema in interstitial tissue of testis were observed in male Wistar rats exposed to mercury (mercuric chloride; 1 mg/kg bw per day; per os–gavage; 4 weeks) [176]. However, it is noteworthy that no histological changes were observed in the testes, neither in Leydig cells nor in seminiferous tubules in Wistar rats receiving tap water containing methylmercury [177].

An effect of a very low dose of mercury (4 ppm) on possible induction of testicular damage was examined also in mice. Mercury (drinking water; CD-1 male mice; 4 ppm HgCl_2_; 12 weeks) significantly reduced the epididymal spermatozoa number. Histological study showed that mercury caused degenerative lesions in the testes [178]. Authors also report that mercury exposure leads to disruption of spermiation as disintegration and necrosis of spermatocytes were detected.

A significant decrease in the total volume of testes, diameters of seminiferous tubules, and total volume of seminiferous tubules was observed in rats exposed to mercury vapor at 1 mg/m^−3^ per day in a chamber for six weeks [179]. A significant decrease was also detected in the numbers of Sertoli cells, spermatogonia, spermatocytes, and spermatids. The spermatogenic cells were degenerated, and seminiferous tubules were atrophied [179].

In another study, testicular atrophy was induced by mercuric chloride [180]. The formation of a fibrotic histopathological structure of mature active seminiferous tubules was seen in the testes of rats exposed to mercury via s.c. injection (5 mg/kg mercury chloride; 5 days). The mercury experimental group also showed a decreased number of spermatocytes [180]. Table 6 summarizes the most significant changes in testes due to mercury exposure.

One study aimed to assess the effects and underlying mechanisms of chronic exposure to low levels of mercury (Wistar rats; HgCl_2_; i.m.; 1st dose 4.6 µg/kg and subsequent doses 0.07 µg/kg per day; 60 days) [177]. Administration of mercury decreased spermatozoa production, count, and motility and increased head and tail morphologic abnormalities. Within head phenotypes, more banana head and total head abnormalities were observed. For tail morphology, more bent tail and total tail abnormalities were detected. In the spermatozoa, motility-decreased type A spermatozoa accompanied by increases in type C spermatozoa (immotile) in mercury-treated rats were observed [178]. In a subsequent paper, authors proved the correlation between sperm pathologies and enhanced oxidative stress in reproductive organs. The authors concluded that chronic exposure to low mercury doses impairs spermatozoa quality and adversely affects male reproductive functions, which may be due to enhanced oxidative stress [183].

In rats, mice, hamsters, and guinea pigs exposed to mercury in the form of mercuric chloride intraperitoneally (1, 2 or 5 mg/kg; 1 month), testicular degeneration and cellular deformation of the seminiferous tubules and Leydig cells were found at the highest dosage. On the other hand, a lower dose of mercury resulted only in testicular degeneration in hamsters and partially in rats and mice with no effects in guinea pigs [184,185].

A reproductive alteration due to mercury was also reported in male Sprague Dawley rats (1.23 mg/kg/bw, 7 days). Authors confirm edema in the interstitial areas, necrosis of spermatogonia, degeneration, edema in intertubular areas, severe thinning of the tubular wall, and a higher percentage of dead spermatozoa in semen [183].

Furthermore, an effect mercury on the motility and structural integrity of rabbit spermatozoa has been demonstrated using an in vitro cell-culture study (5.0–83.3 µg HgCl_2_/mL) [184]. Decreased spermatozoa motility in mercury-exposed cultures was found. Detailed analysis showed a decrease in spermatozoa distance and velocity parameters, straightness, linearity, wobble amplitude of lateral head displacement, and beat cross frequency of spermatozoa. In mercury-exposed cultures, a positive reaction proved alteration in the anterial part of the head (acrosome), connection part (connection piece), and mitochondrial segment [185].

## 5. Conclusions and Future Directions

The signs of toxicity of cadmium, lead, and mercury in reproductive organs appear to be strikingly similar when each is administered individually. In ovaries, the most significant changes are decreased follicular growth, increased number of atretic follicles, degeneration of the corpus luteum, and prolonged and/or irregular cycle. In testes, the most significant changes include disorganization of seminiferous tubules; alterations in spermatogenic cell arrangement; alterations in the basal membrane structure; abnormalities of the testicular stroma; decreased spermatozoa count, motility, and viability; and altered spermatozoa morphology. These are signs of adverse effects of cadmium, lead, and mercury on the architecture of reproductive organs, which are both dose- and time-dependent. In general, toxic effects of various substances in reproductive organs occur at low concentrations. Because toxic mechanisms of each individual metal have been established, future research should be aimed to elucidate molecular mechanism(s) of action of these metals in combinations to mimic human co-exposure situations. In addition, toxicity preventive strategies and the synergistic or antagonistic interactions during the simultaneous presence of more than one of these three metals should be examined in future research.

## Figures and Tables

**Table 1 toxics-08-00094-t001:** Alteration in ovaries induced by cadmium.

Administration/Dose/Species/Form	Changes	References
**Per os**	- Higher number of antral and atretic follicles	Ruslee et al., 2020 [39]
5 mg/kg; 6 weeks, daily; 6 weeks		
Sprague Dawley rats		
CdCl_2_		
**Single i.p.**	- Decrease in the relative volume of growing follicles; - Increased number of atretic follicles	Massanyi et al., 2020 [37]
1.5 mg/kg; killed after 48 h		
**Per os**		
1.0 mg/kg/day; 5 months		
Rabbits		
CdCl_2_		
**Per os (gavage)**	- Prolongation of the cycle length - Degeneration of the corpus luteum - Damaged and fewer oocytes	Nasiadek et al., 2019 [40]
0.09–4.5 mg/kg, 90 days		
Rat (Wistar)		
CdCl_2_		
**Per os**	- Decrease in follicle number	Nna et al., 2017 [41]
5 mg/kg; 14 day		
Rat (Wistar)		
CdCl_2_		
**Per os**	- Increased numbers of atretic follicles - Decreased number of follicles in different stages of maturation - Disorganization, edema and decreased number of yellow bodies	Lubo-Palma et al., 2006 [52]
50, 100 and 150 ppm (in water); 50 days		
Swiss albino mice		
CdCl_2_		

i.p.—intraperitoneal administration.

**Table 2 toxics-08-00094-t002:** Alteration in testes induced by cadmium.

Administration/Dose/Species/Form	Changes	References
**i.p.**	- Decreased number of spermatogenic tubules - Decreased cell level in the spermatogenic tubules - Disordered spermatogenic cells	Han et al., 2020 [67]
2.5 mg/kg; 35 days		
Kunming mice		
CdCl_2_		
**i.p.**	- Alterations in spermatogenic cells arrangement - Irregular layers of seminiferous tubes - Loss of spermatogenic cells and spermatozoa	Liu et al., 2020 [60]
3 mg/kg; 5 days and 1 mg/kg for 30 days		
BALB/c mice		
CdCl_2_		
**Per os**	- Decreased spermatozoa count, motility and viability; altered spermatozoa morphology - Atrophy of the seminiferous tubules - Disrupted testicular architecture	Olaniyi et al., 2020 [68]
5 mg/kg, 30 days		
Rats (Wistar)		
CdCl_2_		
**Per os**	- Thin germinal epithelium, seminiferous tubules with aberrant morphology - Markedly low level of normal spermatogenesis - Abnormalities of the testicular stroma	Ren et al., 2019 [65]
2, 4, 8 mg/kg, 8 days		
Mice (Institute of Cancer Research male specific pathogen-free)		
CdCl_2_		
**i.p.**	- Significant decrease in germinal epithelium volume and increase in stroma volume - Various injury of the seminiferous epithelium - Alterations in the basal membrane structure	Toman et al., 2002 [69]
single dose, 2.25 mg/kg, 48 h		
**Per os**		
1.0 mg Cd/kg, 5 months		
Rabbit		
CdCl_2_		

i.p.—intraperitoneal administration.

**Table 3 toxics-08-00094-t003:** Alteration in ovaries induced by lead.

Administration/Dose/Species/Form	Changes	References
**Per os**	- Follicular edema - Ovarian follicle necrosis	Uchewa and Ezugworie, 2019 [100]
1.5 mg/kg daily; 21 days		
Wistar rat		
Lead acetate		
**Per os (drinking water)**	- Areas with optical empty spaces - Diffuse edemas and ovarian follicle denudation - Necrosis of the ovarian follicles	Dumitrescu et al., 2015 [95]
0.050–0.150 mg/L; 12 months		
Wistar female rats		
Lead acetate		
**Per os (gavage)**	- Decreased primary follicular count - Interference with the development of growing follicles in the ovary	Waseem et al., 2014 [93]
30 mg/kg/day; two months		
mice BALBc		
Lead acetate		
**Per os**	- Irregular estrous cycle; drop of fertility rate - Atresia in all the stages of folliculogenesis	Dhir and Dhand, 2010 [94]
60 mg/kg; 90 days		
rats (Disease-free albino rats)		
Not specified		
**i.p.**	- Dysfunction of folliculogenesis - The deceased amount of primordial follicles - Increase in atretic antral follicles	Taupeau et al., 2001 [92]
acute 10 mg/kg; 15 days; chronic 10 mg/kg; 15 weeks		
Mice, C57 Bl × CBA		
Pb(NO_3_)		

i.p.—intraperitoneal administration.

**Table 4 toxics-08-00094-t004:** Alteration in testes induced by lead.

Administration/Dose/Species/Form	Changes	References
**Per os**	- Disorganization of seminiferous tubules - Complete absences of the spermatogenesis	Elsheikh et al., 2020 [111]
100 mg/kg; 3 weeks		
Kunming mice		
Lead acetate		
**Per os**	- Degeneration of testicular tissue with loss of spermatogenic series - Elevation of ROS level, lipid peroxide levels and lysosomal enzyme activity	Kelainy et al., 2019 [113]
20 mg/kg; daily for 10 days		
Albino rats		
Lead acetate		
**Per os**	- Edema, hydrocele and inflamed tunica albuginea	Ezejiofor and Orisakwe, 2019 [114]
50 g of lead acetate dissolved in 12 mL of 1N HCl; 4 weeks		
Albino Wistar rats		
Lead acetate		
**Per os**	- Significant decrease in the weights of testes - Marked testicular lesions of seminiferous tubules - Disorganization seminiferous tubules, complete hyalinization, tubular blockage, sloughed germinal epithelium, and germinal epithelium hypocellularity - Decreased spermatogenesis score	Hassan et al., 2019 [116]
20 mg/kg, 56 days		
Albino rats		
Lead acetate		
**Per os**	- Significant reduction in testis weight, spermatozoa count, testosterone levels and, antioxidant enzymes levels - Devoid of germ cells and maturation arrest; formation of giant primary spermatocytes	Santhoshkumar and Asha Devi, 2019 [117]
Wistar rats		
0.15%; 45 days		
Lead acetate		

**Table 5 toxics-08-00094-t005:** Alteration in ovaries induced by mercury.

Administration/Species/Form	Changes	References
**i.m.**	- Irregular estrous cycles - Abnormal duration spent in the different phases of the estrous cycle - Reduced number of ovarian antral follicles number - Increase in the number of atretic ovarian follicles - Increase of lipid deposition	Merlo et al., 2019 [158]
4.6 μg/kg + subsequent dose 0.07 μg/kg; 30 days		
Wistar rats		
Mercuric chloride		
**inhalation**	- Necropsy of corpora lutea at estrus or metestrus - Immature corpora lutea - Prolonged estrous cycles	Davis et al., 2001 [153]
1–4 mg/m^3^ Hg°; 11 days		
Sprague–Dawley rats		
Hg° vapor		
**s.c.**	- Retarded follicular development and morphologically prolonged corpora lutea	Lamperti et al., 1973 [163]
1 mg per day; each day of the 4-day cycle,		
Golden hamsters		
Mercuric chloride		
**s.c.**	- Higher mercury concentration in the corpora lutea than the follicles of the interstitium - Absent ovulation by Day 1 of the third cycle - Atretic follicles in the primary and secondary stages	Lamperti and Printz, 1974 [157]
1–4 mg		
Golden hamster		
Mercuric chloride		

i.m.—intramuscular administration; s.c.—subcutaneous administration.

**Table 6 toxics-08-00094-t006:** Alteration in testes induced by mercury.

Administration/Species/Form	Changes	References
**Per os (gastric gavage)**	- Degeneration of the spermatogenic cells of the germinal epithelium - Occlusion of the lumen of seminiferous tubules - Hypertrophy of seminiferous tubules - Irregular vacuolized basement membrane	Adelakun et al., 2020 [174]
40 mg/kg; once a day; 28 consecutive days		
Wistar rats		
Mercuric chloride		
**s.c.**	- Formation of fibrotic histopathological structure of mature active seminiferous tubules - Decreased number of spermatocytes	Fadda et al., 2020 [180]
5 mg/kg; 5 days		
Wistar rats		
Mercury chloride		
**i.p.**	- Severe edema in the interstitium - Necrotic and degenerative changes - Thinned tubular wall - Severe levels of TNF-α and COX-2 expressions in the intertubular areas	Kandemir et al., 2020 [179]
1.23 mg/kg; 7 days		
Sprague Dawley rats		
Mercuric chloride		
**Per os**	- Decline in spermatozoa - Disorganization and degeneration of spermatogenic cells - Vacuolated area within the seminiferous tubules	Almeer et al., 2020 [181]
0.4 mg/kg; 28 days		
Wistar rats		
Mercuric chloride		
**i.p. (single)**	- Undulation of basal membrane - Dilatation of blood vessels in interstitium - Occurrence of empty spaces in germinal epithelium - Decreased relative volume of germinal epithelium, increased relative volume of interstitium - Increased apoptosis - Decreased number of nuclei in germinal epithelium	Massányi et al., 2007 [182]
5–20 mg/kg		
Rats		
Mercuric chloride		

s.c.—subcutaneous administration; i.p.—intraperitoneal administration.

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
