# Peer review of "Effects of Cadmium, Lead, and Mercury on the Structure and Function of Reproductive Organs"

_toxics, 2020, doi:10.3390/toxics8040094_

Round 1
Reviewer 1 Report
Reviewer’s comments
General comments
The review manuscript by Massanyi P. et al. (toxics-922771) deals with the effect of toxic metals (cadmium, lead, mercury) on fertility. The subject of the review is quite interesting. The authors do not mention, neither in the introduction nor in the abstract, the novelty of the specific review, as well as whether there are any pertinent or relevant published reviews. The structure of the paragraphs in the 3 metals examined is common, demonstrating experimental results. However, the information provided, excluding the tables, are neither combined nor evaluated as a whole. The text could be modified, adding a final section, where the most significant information would be compiled, in order to assist the reader. The manuscript could be published, provided it is appropriately reconsidered.
Detailed comments
- The review refers almost exclusively to the effects of Cd, Pb and Hg on fertility. The title could be modified to ‘Cadmium, lead and mercury effects on structure and function of reproductive organs’.
- In keywords, replace ‘trace elements’ with ‘trace metals’.
- No data are presented regarding the synergistic or antagonistic interactions during the simultaneous presence of more than one of the three metals examined.
- In recent years a gradual decrease of mainly Pb concentrations in the environment is recorded. Are there any data referring to the fertility of human population, in relation to the specific reduction of metal levels? In addition, in l.20 (abstract) the sentence ‘The pollution of environmental increase all over the world’ should be reconsidered.
- In Tables 1, 2, 3 the abbreviations particularly in the second column (Application/Dose/Species/Form) should be explained.
Author Response
Authors thank the Reviewer for all comment and suggestions. All corrections and changes in the text are in blue letters.
Point 1: The authors do not mention, neither in the introduction nor in the abstract, the novelty of the specific review, as well as whether there are any pertinent or relevant published reviews.
Response: We have rewritten the abstract and introduction to address the reviewer’s concern. The novelty of our work is a comprehensive review of adverse effects on reproductive organs of three toxic elements, cadmium, lead and mercury. These three toxic elements have been chosen based on the WHO list of ten chemicals of major public health concern (https://www.who.int/ipcs/assessment/public_health/chemicals_phc/en/). In addition, our review focuses on adverse effects of these three toxic elements on the structure and function of reproductive organs evident from microscopic investigations. We believe that these aspects have not been covered thoroughly in any previous published reports. This is reflected by our search attempt (cadmium AND lead AND mercury AND testis AND ovary AND review) that gave us the result: PUBMED: 1; WOS: 1; SCOPUS: 4.
Point 2: The structure of the paragraphs in the 3 metals examined is common, demonstrating experimental results. However, the information provided, excluding the tables, are neither combined nor evaluated as a whole.
Response: We have now added a new section 5. Conclusion and future direction (lines 650-664).
Point 3: The text could be modified, adding a final section, where the most significant information would be compiled, in order to assist the reader.
Response: We summarize findings in Tables 1 and 2 for cadmium, tables 3 and 4 for lead and tables 5 and 6 for mercury.
Detailed comments
Point 4: The review refers almost exclusively to the effects of Cd, Pb and Hg on fertility. The title could be modified to ‘Cadmium, lead and mercury effects on structure and function of reproductive organs’
Response: To better reflect the content of our review, as advised, the title has now been changed to read, “Effects of Cadmium, Lead and Mercury on the Structure and Function of Reproductive Organs”.
Point 5: In keywords, replace ‘trace elements’ with ‘trace metals’
Response: A correction has been undertaken.
Point 6: No data are presented regarding the synergistic or antagonistic interactions during the simultaneous presence of more than one of the three metals examined.
Response: We agree that the interactive effects of metals in combination are relevant to real life situations. However, to address the interactions, knowledge on effects of each individual metal must be demonstrated first. As stated in response to Point 1, our paper aims to make existing evidence for effects of individual metals apparent. We believe that these data should pave way to further study using various combinations that would closely mimic human exposure situations. The reviewer’s suggestions have been inserted in Section 5: Conclusion and future direction (lines 660-664).
Point 7: In recent years a gradual decrease of mainly Pb concentrations in the environment is recorded. Are there any data referring to the fertility of human population, in relation to the specific reduction of metal levels? In addition, in l.20 (abstract) the sentence ‘The pollution of environmental increase all over the world’ should be reconsidered.
Response: We agree with the Reviewer that there is evidence of a reduction in Pb in the environment. However, there is little evidence for that trend for Cd. We have deleted the referred sentence in the abstract.
Point 8: In Tables 1, 2, 3 the abbreviations particularly in the second column (Application/Dose/Species/Form) should be explained.
Response: Abbreviations are explained. The form of application is in bold.

Reviewer 2 Report
It was my pleasure to peer review the manuscript entitled ‘Exposure to toxic metals and their effects on fertility’. The manuscript tries to characterise exposure to cadmium, lead and mercury and their reproductive toxicity. The topic is interesting; however, the manuscript does not have a clear focus neither on ‘exposure’ nor ‘fertility’, thus should be greatly improved in order to be considered for publication.
Point 1. Authors should consult language editing service to improve their English writing.
Point 2. There are many toxic metals, but manuscript only picks up cadmium, lead and mercury. The manuscript title should be re-considered.
Point 3. There are many review articles published previously that summarises reproductive toxicity of cadmium, lead and mercury. Authors should explain in the introduction what is new in this manuscript.
Point 4. The title says ‘exposure to toxic metals’ but no exposure levels are presented in the manuscript. Thus, it gives no relevance to human risk assessment.
Point 5. If human fertility is to be discussed, authors should consider more essential factors such as fecundity, time to pregnancy, treatment rate and semen quality.
Author Response
Authors thank reviewer for all comment and suggestions. All corrections and changes in the text are in blue letters.
It was my pleasure to peer
review the manuscript entitled ‘Exposure to toxic metals and their effects on fertility’. The manuscript tries to characterise exposure to cadmium, lead and mercury and their reproductive toxicity. The topic is interesting; however, the manuscript does not have a clear focus neither on ‘exposure’ nor ‘fertility’, thus should be greatly improved in order to be considered for publication.
Point 1: Authors should consult language editing service to improve their English writing.
Response: Our paper has been subjected the English editing (MDPI's English editing service).
Point 2: There are many toxic metals, but manuscript only picks up cadmium, lead and mercury. The manuscript title should be re-considered.
Response: The title of our review has been changed to, “Effects of Cadmium, Lead and Mercury on the Structure and Function of Reproductive Organs”.
Point 3: There are many review articles published previously that summarises reproductive toxicity of cadmium, lead and mercury. Authors should explain in the introduction what is new in this manuscript.
Response: We have rewritten the introduction and abstract to address the reviewer’s concern. The novelty of our work is a comprehensive review of adverse effects on reproductive organs of three toxic elements, cadmium, lead and mercury. These three toxic elements have been chosen based on the WHO list of ten chemicals of major public health concern (https://www.who.int/ipcs/assessment/public_health/chemicals_phc/en/). In addition, our review focuses on adverse effects of these three toxic elements on the structure and function of reproductive organs evident from microscopic investigations. We believe that these aspects have not been covered thoroughly in any previous published reports. This is reflected by our search attempt (cadmium AND lead AND mercury AND testis AND ovary AND review) that gave us the result: PUBMED: 1; WOS: 1; SCOPUS: 4.
Point 4: The title says ‘exposure to toxic metals’ but no exposure levels are presented in the manuscript. Thus, it gives no relevance to human risk assessment.
Response: We have provided in Tables 1-6, the administered doses of cadmium, lead and mercury that produced adverse effects on the reproductive organs (testes, ovaries).
Point 5: If human fertility is to be discussed, authors should consider more essential factors such as fecundity, time to pregnancy, treatment rate and semen quality.
Response: We agree with the Reviewer that developmental toxicity should become a prominent topic if fertility is to be discussed. However, in the present review, we describe toxic effects seen in reproductive organs due to cadmium, lead and mercury for the reason that has been stated in response to Point 3 above. In an attempt to clarify aspects covered in our review, we have rewritten abstract and part of introduction. We also have added a new Section 5: Conclusion and future direction (lines 650-664).

Reviewer 3 Report
This Ms aims to explore the adverse effects of heavy metals on reproductive organs of animals and humans. The title does not reflect the content of the MS. Aims of study are not clear. The English language is very poor which makes it difficult to understand the text. The first sentence in the abstract in incorrect. For example testis and ovary are relevant reproductive and hormonal organs, with implication in multiple body functions.
A reorganization of all text and tables is suggested. For example, each Table should include, in different columns, the species under study, doses and form of exposure, organs / tissues studied, main results and references.
It is suggested that in the end the authors point out lines of future research. Please select references.
Other comments:
L.15, and 34 - Please rephrase the sentence: “Reproductive organs are not … development of species.
L.43 – Which is the meaning of this sentence? Development of industry and agricultural production leads to reorganization of elements in the food chain.
L.240 – It is obvious “Young children are more vulnerable to toxicity of lead”.
Author Response
Authors thank reviewer for all comment and suggestions. All corrections are highlighted in the text (blue).
Point 1: This Ms aims to explore the adverse effects of heavy metals on reproductive organs of animals and humans. The title does not reflect the content of the MS.
Response: To match with its content, as advised, the title has been changed to, “Effects of Cadmium, Lead and Mercury on the Structure and Function of Reproductive Organs”.
Point 2: Aims of study are not clear. The English language is very poor which makes it difficult to understand the text. The first sentence in the abstract in incorrect. For example, testis and ovary are relevant reproductive and hormonal organs, with implication in multiple body functions.
Response: The abstract has been rewritten. A clear statement on aim of our review is included as quoted below. Our paper has been subjected the English editing (MDPI's English editing service). We have corrected the first sentence.
“The present review aims to provide results from experimental studies demonstrating toxic effects of cadmium, lead and mercury on the structure and function of reproductive organs and the administered doses at which toxic effects are observed.”
Point 3: A reorganization of all text and tables is suggested. For example, each Table should include, in different columns, the species under study, doses and form of exposure, organs / tissues studied, main results and references.
Response: As advised, we have reorganized Tables and their entries, including doses and form of exposure, organs / tissues studied, main results and references.
Point 4: It is suggested that in the end the authors point out lines of future research. Please select references.
Response: We have provided some suggestion for future research in a new Section 5: Conclusion and future direction (lines 650-664).
Other comments:
L.15, and 34 - Please rephrase the sentence: “Reproductive organs are not … development of species.
Response: The referred statement in the abstract has been changed to read as “Reproductive organs are essential not only for the life of an individual but also for the survival and development of species.”
L.43 – Which is the meaning of this sentence? Development of industry and agricultural production leads to reorganization of elements in the food chain.
Response: We have rewritten the Introduction and we have changed the referred statement to read as quoted below (lines 54-55).
“The industrial development and agricultural activities have resulted in varying degrees of environmental pollution and reorganization of toxic elements in the food chain.”
L.240 – It is obvious “Young children are more vulnerable to toxicity of lead”.
Response: The referred statement has been deleted.

Round 2
Reviewer 1 Report
In the revised manuscript toxics-922771 the authors have dealt with the comments.
Author Response
Reviewer 1
Authors thank the Reviewer for all comment and suggestions.
The reviewer states that “In the revised manuscript toxics-922771 the authors have dealt with the comments”.
This is a great appreciation of our hard work ?
Reviewer 3 Report
Authors have done several changes to immprove the MS. However, please find the following comments:
Abstract & Introduction sections
The aims of study should include if authors refers to humans or other mammals.
L.48-51 – Please rephrase, since there are some studies already published:
doi: 10.1002/jat.3122.
doi: 10.5604/12321966.1152077.
doi: 10.4103/1008-682X.150847.
doi: 10.1016/j.chemosphere.2015.10.078.
doi: 10.1089/dna.2017.4081.
L.524, 541 - in ovary
L.660 – Please rephrase “the molecular mechanism(s) of action of each individual metal…” since it is already konwn.
Author Response
Authors thank reviewer for all comment and suggestions. We provide point-by-point response to the Reviewer’s comments as below. Changes made to the text are highlighted.
Point 1: Abstract & Introduction sections
The aims of study should include if authors refer to humans or other mammals.
Response: As advised, we have rewritten abstract and introduction to reflect that our review covers studies in humans and animals that included mammalian and avian species as quoted below).
“The present review focused on experimental studies using rats, mice, avian and rabbits to demonstrate unambiguously effects of cadmium, lead or mercury on the structure and function of reproductive organs. In addition, relevant human studies are discussed.”
In addition to statements in abstract, we have inserted latest data on human exposure to mercury in foods in the text (lines 457-466) and references (146-148) as quoted below.
“In an analysis of data from the U.S. National Health and Nutrition Examination Survey (NHANES) 2011-2012 (N = 7920), the overall population mean for whole blood total mercury (THg) was 0.70 μg/L [146]. In a comparative study, 3.8% of seafood consumers had whole blood THg higher than 5.8 μg/L, while 9.4% of them had whole blood THg higher than 3.4 μg/L [147]. In addition, seafood consumers had higher geometric mean for whole blood THg (0.89 µg/L) than non-seafood consumers (0.31 µg/L) [147]. Fish/seafood was found to be likely sources of mercury exposure among seafood consumers, whereas wine, rice, vegetables, vegetable oil, or liquor were dietary sources of mercury exposure among non-seafood consumers [147]. Intriguingly, a 2.57-fold increase in risk of infertility was seen among women enrolled in NHANES 2013-2016 who had high levels of whole blood THg (>5.278 μg/L) [148].”
- Mortensen, M.E.; Caudill, S.P.; Caldwell, K.L.; Ward, C.D.; Jones, R.L. Total and methyl mercury in whole blood measured for the first time in the U.S. population: NHANES 2011-2012. Environ. Res. 2014, 134, 257-264.
- Wells, E.M.; Kopylev, L.; Nachman, R.; Radke, E.G.; Segal, D. Seafood, wine, rice, vegetables, and other food items associated with mercury biomarkers among seafood and non-seafood consumers: NHANES 2011-2012. J. Exp. Sci. Environ. Epidemiol. 2020, 30, 504-514.
- Zhu, F.; Chen, C.; Zhang Y.; Chen, S.; Huang, X.; Li, J.; Wang, Y.; Liu, X.; Deng, G.; Gao, J. Elevated blood mercury level has a non-linear association with infertility in U.S. women: Data from the NHANES 2013-2016. Reprod. Toxicol. 2020, 91, 53-58.
Point 2: L.48-51 – Please rephrase, since there are some studies already published:
Response: We have rephrased the referred statements (lines 66-67) and all five references suggested by the reviewers have been cited (references 4-9).
“Accordingly, we sought to review evidence for reproductive toxicity induced by cadmium, lead and mercury focusing on changes in the structure and function of male and female reproductive organs of various animal species. Some of these aspects have been covered in published reports [4-9].”
Point 3: L.524, 541 - in ovary
Response: A correction has been undertaken.
Point 4: L.660 – Please rephrase “the molecular mechanism(s) of action of each individual metal…” since it is already known.
Response: As suggested, we have rephrased the referred statement to read as below. Changes in the text are highlighted (lines 670-672).
“Because toxic mechanisms of each individual metal have been established, future research should be aimed to elucidate molecular mechanism(s) of action of these metals in combinations to mimic human co-exposure situations.”